# A Green-emitting Fluorescent Probe Based on a Benzothiazole Derivative for Imaging Biothiols in Living Cells

**DOI:** 10.3390/molecules24030411

**Published:** 2019-01-23

**Authors:** Xiaohua Ma, Yuanqiang Hao, Jiaxiang Liu, Guoguang Wu, Lin Liu

**Affiliations:** 1School of Chemical Engineering and Technology, China University of Mining and Technology, Xuzhou 221116, China; maxiaohua@sqnu.edu.cn; 2Henan Key Laboratory of Biomolecular Recognition and Sensing, College of Chemistry and Chemical Engineering, Shangqiu Normal University, Shangqiu 476000, China; liujiaxiang2015@aliyun.com; 3Key Laboratory of New Optoelectronic Functional Materials (Henan Province), College of Chemistry and Chemical Engineering, Anyang Normal University, Anyang 455000, China

**Keywords:** fluorescent probe, large Stokes shift, biothiols, cell imaging, green-emitting

## Abstract

A new green-emitting fluorescent probe **1** was developed for biothiol detection. The sensing mechanism was considered to be biothiol-induced cleavage of the 2,4-dinitrobenzene- sulfonate group in probe **1** and resulting inhibition of the probe’s photoinduced electron transfer (PET) process. Probe **1** exhibited favorable properties such as excellent selectivity, highly sensitive (0.12 µM), large Stokes shift (117 nm) and a remarkable turn-on fluorescence signal (148-fold). Furthermore, confocal fluorescence imaging indicated that probe **1** was membrane-permeable and suitable for visualization of biothiols in living A549 cells.

## 1. Introduction

Biological thiols (biothiols) play essential functions in many physiological processes of prokaryotes and eukaryotes owing to their redox-relevant role and nucleophilicity [1,2,3]. Among them, cysteine (Cys), homocysteine (Hcy) and glutathione are the most plentiful low molecular weight and typical endogenous biothiols [4,5,6]. Their content is of critical significance, and the abnormal levels of biothiols are directly linked with the occurence of some diseases. For example, Cys is an essential amino acid for the production of proteins; Cys deficiency can result in hair depigmentation, children slow growth, liver damage and edema. Homocysteine has the function of detoxification and metabolism and heightened level of Hcy in plasma is considered as a risk factor for incidence of cardiovascular and Alzheimer’s diseases. GSH is known as maintaining normal immune system function and helping antiaging, while its abnormal levels are associated with cancer and HIV infection [7,8,9,10,11,12,13,14,15]. At present, there are a series of analytical methods for thiol detection in the clinic, such as high-performance liquid chromatography (HPLC), electrochemical methods and mass spectrometry [16,17,18]. Artificial sensors for monitoring the distribution and concentration of biothiols in biological samples have always attracted considerable attention. In terms of overcoming practical limitations, the use of fluorescence imaging provides a more desirable method due to its simplicity, great sensitivity, excellent selectivity, non-destructive analysis, as well as portability to real-time detection in actual living systems [19,20,21,22,23].

So far, many fluorescent probes for thiols have been already designed and developed. These studies have greatly advanced the research on the fluorescent detection of thiols [24,25,26,27,28,29,30,31,32,33,34,35,36,37,38]. However, many of these probes show low sensitivity and small Stokes shifts, which limits their practical applications in biological systems. It is known that a large Stokes shift is beneficial for avoiding measurement errors derived from the self-quenching and autofluorescence [39,40]. Therefore, the development of an efficient fluorescent probe for the accurate detection of biothiols is still highly desired. 

Due to its good photostability, large Stokes shift, good cell membrane permeability and low toxicity, 2-(benzothiazol-2-yl)phenol and its derivatives have been widely used for the design of fluorescent reagents [41,42,43]. Herein, we identified the 2,4-dinitrobenzenesulfonate moiety (DNBS) as a reactive group and reaction switch specific for biothiols. Integration of this unique moiety into the phenothiazine benzothiazole-based dye **2** fluorophore afforded a new turn-on fluorescence probe **1** for detection of biothiols (Scheme 1). In view of the recommendable properties such as highly selective, remarkable fluorescence turn-on signal (148-fold enhancement), relatively low detection limit (0.12 µM) and a 117 nm Stokes shift, probe **1** has been successfully utilized to image biothiols in living A549 cells.

## 2. Results and Discussion

### 2.1. Spectroscopic Studies

The synthesis and characterization of probe **1** are described in the Materials and Methods section and in the Appendix A (Appendix A). We firstly evaluated the sensing capability of probe **1** to detect biothiols in PBS buffer solution (50.0 mM, pH 7.4, 20% DMSO). Probe **1** showed a maximum absorption peak centering at 384 nm and little fluorescence due to an efficient photo induced electron transfer (PET) process [44] from the dye **2** to the 2,4-dinitrobenzenesulfonate moiety (Figure 1). 

After incubation with Cys (100.0 μM) for 15 min, a strong fluorescence (Figure 1B) at 530 nm was observed. Under these conditions, dye **2** displayed an absorption centred at 413 nm and an emission with λ_max_ = 530 nm (Figure 1). Detection of compound 2 by HRMS analysis demonstrated that the sensing mechanism was the sulfonate ester cleavage mediated by biothiols in probe **1** to generate the dye **2** as shown in Scheme 1. Moreover, the HRMS spectral analysis of probe **1** treated with Cys confirmed the formation of dye **2**. A prominent peak at m/z 405.1105 corresponding to the dye **2** was clearly observed (Appendix A).

### 2.2. Sensitivity Studies 

We tested the sensitivity of probe **1** (5.0 μM) toward Cys. Upon addition of Cys (0.0~100.0 μM), fluorescent intensity at 530 nm gradually increase with the concentration of Cys. The change in fluorescent titration spectra terminated when the concentration of Cys reached 100.0 µM. At such concentration, the fluorescence intensity became about 148-fold higher than the baseline with no Cys (Figure 2). 

As seen in Figure 3, a good linear relationship (*y* = 12.5969 + 20.2985*x*, R^2^ = 0.9972) between fluorescence intensity at 530 nm and the Cys concentrations (0.0~10 µM) was obtained. The detection limit of probe **1** for Cys was calculated to be 0.12 μM on the basis of S/N = 3. Additionally, similar results were obtained affording 0.14 μM and 0.21 μM detection limits for Hcy and GSH, respectively (Appendix A). The results demonstrated that probe **1** could be used as a turn-on fluorescent indicator for biothiols in aqueous media. 

### 2.3. Selective and Interference Fluorescence Response to Thiols

To investigate the selectivity performance of probe **1** (5.0 μM) for biothiols, biologically relevant compounds (100.0 μM), including Ala, Asn, Arg, Asp, Gly, Gln, Glu, Lys, Leu, Met, Phe, Pro, Trp, Tyr, Thr, Ser, His, Ile, Val, GSH, Hcy, Cys, KCl, NaCl, MgCl_2_, CaCl_2_, ZnCl_2_, FeCl_3_, Na_2_SO_4_, NaSCN, AcONa, Na_2_CO_3_, NaNO_3_, Na_3_PO_4_, NaNO_2_ and H_2_O_2_ were examined. As a result, only biothiols (Cys, GSH and Hcy) elicited a significant fluorescence enhancement of probe **1**, whereas representative amino acids, typical biological ions and molecules had no response towards probe **1** (Figure 4A and Appendix A). Moreover, competitive assays indicated that probe **1** still retained sensing biothiols in the presence of representative amino acids, typical biological ions and molecules (Figure 4B and Appendix A). These phenomena confirmed that probe **1** had highly selective for biothiols over other competitive species under physiological conditions.

### 2.4. Kinetics and the Effects of pH

The real-time reaction kinetics of probe **1** towards biothiols was performed. As displayed in Figure 5A, upon the addition of biothiols (100.0 μM for Cys, GSH, Hcy), a high-intensity green emission at 530 nm was observed (148, 127 and 102 folds enhancement after 15 min, respectively). It was attributed to the cleavage of 2,4-dinitrobenzenesulfonate moiety in probe **1,** thereby eliminating the PET process to generate the dye **2**. However, the free probe **1** brought about negligible fluorescence change with the reaction time prolonged. These results manifested that the newly developed probe **1** possessed fast response characteristic toward biothiols. We proceeded to investigate the effect of pH on the response of probe **1** to biothiols (Cys). As seen in Figure 5B, free probe **1** (5.0 μM) exhibited no significant fluorescence fluctuation at 530 nm in broad pH range from 2.0 to 12.0, suggesting that the probe **1** works steadily in this pH range. However, addition of biothiols (100.0 μM) induced remarkable fluorescence enhancement at 530 nm in the pH range of 5.0~9.0. This indicated that the probe **1** was independent of pH and properly favorable for monitoring biothiols in biological environment applications.

### 2.5. Bioimaging of Probe ***1***

A549 cells were taken for monitoring the practical applicability of probe **1** to detect biothiols in live cells. The pictures of brightfield and fluorescence images were performed through confocal fluorescence microscopy. As is depicted in Figure 6, the A549 cells were stained with probe **1** solely (5.0 μM), and green fluorescence signal was appeared inside the cells (d, e and f). In sharp contrast, when the A549 cells were pretreated with NEM (1.0 mM) for 30 min to diminish the interference from intracellular biothiols, and then subsequently stained with probe **1**, there was almost no remarkable fluorescence after the above treatment (a, b and c). The capability of probe **1** for imaging biothiols in Hela cells was also evaluated, and the results displayed a similar pattern with that of A549 cells (Appendix A). Cellular viability tests further demonstrated that probe **1** is highly biocompatible (Appendix A). These results manifested that probe **1** was cell-permeable and suitable for rapid visualization of biothiols in biological applications.

## 3. Materials and Methods

### 3.1. Instruments and Chemicals

The data of absorption spectra and emission spectra were carried out using UV-2450 and RF5301PC instruments from Shimadzu (Kyoto, Japan). Emission spectra were obtained with a set 5.0 nm for slit widths of excitation and emission. Cell imaging, mass spectrometric experiments, NMR spectra and pH measurements were performed on FV1000 microscope (Olympus, Kyoto, Japan) Xevo G2-S QTof™ mass spectrometer (Waters^®^, Manchester, UK), Bruker 400 spectrometer (Rheinstetten, Germany) and PHS-3C pH meter (Leici, Shanghai, China), respectively. All reagents were commercially available and not further purified to use. TLC plates and mesh 200–300 silica gel were purchased from Qingdao Chemical (Qingdao, China). 

### 3.2. Preparation of Spectra Measurements

The concentration of stock solution of the probe **1** was configured 1.0 mM in DMSO. The tested element (Ala, Glu, Asn, Gly, Arg, Lys, Asp, Gln, Leu, Tyr, Met, Thr, Phe, Pro, His, Trp, Ser, Ile, Val, GSH, Hcy, Cys, KCl, NaCl, MgCl_2_, CaCl_2_, ZnCl_2_, FeCl_3_, Na_2_SO_4_, NaSCN, AcONa, Na_2_CO_3_, NaNO_3_, Na_3_PO_4_, NaNO_2_ and H_2_O_2_) were prepared in twice-distilled water. Place 5.0 μM of probe diluted solution mixed with appropriate analytes stock in 3.0 mL test cell (7.4 PBS buffer containing 20% DMSO), shake well and incubate 15 min each time at room temperature to make spectra measurements.

### 3.3. Cell Assay

A549 cells were seeded overnight to adhere on glass-bottomed dish in DMEM culture medium in air containing 5% CO_2_ at 37 °C. Prior to each experiment, cells were washed with PBS and the culture solution removed. The first group of A549 cell were stained with probe **1** for 30 min, and mounted on the microscope stage to capture the cellular fluorescence images using the Olympus FV1000 microscope. The second group of A549 cell were treated with 1.0 mM N-ethylmaleimide (NEM) prior to staining with the probe **1**. After the above breeding experiment, cells were rinsed with PBS buffer and imaged on confocal microscope immediately.

### 3.4. Synthesis of Dye 2

To a solution of 10-butyl-2-hydroxy-10*H*-phenothiazine-3-carbaldehyde [1] (299 mg, 1.0 mmol) and *o*-aminothiophenol (125 mg, 1.0 mmol) in 10 mL ethanol were added 37% HCl (110 mg, 3.0 mmol) and 30% H_2_O_2_ (204 mg, 6.0 mmol). After stirring at ambient temperature for 2 h, the precipitated solid was collected by filtration and recrystallized (ethanol) to obtain the pure dye **2** (47%). ^1^H-NMR (400 MHz, DMSO-*d_6_*) δ 11.67 (s, 1H), 8.10 (d, *J* = 7.5 Hz, 1H), 8.00 (d, *J* = 7.4 Hz, 1H), 7.85 (s, 1H), 7.51 (s, 1H), 7.41 (d, *J* = 7.4 Hz, 1H), 7.32–7.14 (m, 2H), 7.08 (d, *J* = 7.9 Hz, 1H), 6.99 (d, *J* = 6.9 Hz, 1H), 6.69 (s, 1H), 3.89 (s, 2H), 1.73 (s, 2H), 1.43 (d, *J* = 6.2 Hz, 2H), 0.92 (s, 3H). ^13^C-NMR (100 MHz, DMSO-*d_6_*) δ 164.55, 157.27, 151.94, 148.77, 143.59, 134.47, 128.14, 127.60, 126.85, 126.15, 123.55, 123.47, 122.35, 122.15, 116.68, 104.05, 47.14, 28.54, 19.90, 14.13. HRMS (EI) *m/z* calcd for [C_23_H_20_N_2_OS_2_ + H]^+^: 405.1095, Found: 405.1104.

### 3.5. Synthesis of Probe ***1***

To a mixture of dye **2** (80.0 mg, 0.2 mmol) and Et_3_N (24.0 mg, 0.24 mmol) in dry CH_2_Cl_2_ (10 mL) 2,4-dinitrobenzenesulfonyl chloride (70.0 mg, 0.24 mmol) was added. After stirring at ambient temperature for 7 h, the crude residue was purified by chromatography on silica gel (mixtures of petroleum ether and ethyl acetate as eluent; 6:1, *v*/*v*) to yield the probe **1** (87.1%). ^1^H-NMR (400 MHz, DMSO) δ 8.95 (s, 1H), 8.25 (d, *J* = 14.7 Hz, 2H), 8.06 (d, *J* = 6.8 Hz, 1H), 7.89 (d, *J* = 6.7 Hz, 1H), 7.77 (s, 1H), 7.46 (d, *J* = 31.9 Hz, 2H), 7.34–7.15 (m, 2H), 7.16–6.96 (m, 2H), 6.86 (s, 1H), 3.81 (s, 2H), 1.60 (s, 2H), 1.36 (s, 2H), 0.87 (s, 3H). ^13^C-NMR (100 MHz, DMSO) δ 161.10, 152.72, 151.12, 148.18, 148.04, 146.17, 143.21, 135.34, 133.65, 128.54, 127.88, 127.10, 123.25, 122.53, 121.06, 120.78, 111.02, 47.19, 28.36, 19.79, 13.98. HRMS (EI) *m/z* calcd for [C_29_H_23_N_4_O_7_S_3_ + H]^+^: 635.0729, Found: 635.0737.

## 4. Conclusions

In conclusion, we have developed a new fluorescent off-on probe **1** with high sensitivity and selectivity for biothiol detection. Probe **1** displays remarkable turn-on fluorescence signals (148-fold) at 530 nm and a large Stokes shift (117 nm) towards biothiols. The detection limit of probe **1** for Cys was found to be 0.12 μM in PBS buffer solution. Most importantly, confocal fluorescence imaging of biothiols in living A549 cells suggested that probe **1** might serve as a useful bioanalytical molecular tool.

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
