# Peer review of "A Green-emitting Fluorescent Probe Based on a Benzothiazole Derivative for Imaging Biothiols in Living Cells"

_molecules, 2019, doi:10.3390/molecules24030411_

Round 1
Reviewer 1 Report
Ma et al. describes a turn-on fluorescent probe for detecting thiols inside cells based on a thiol-induced cleavage of 2,4-dinitrobenzenesulfonate from a green-emitting phenothiazine benzothiazole dye. The authors showed that probe 1 can react with biothiols (cysteine, homocysteine, and glutathione) generating a green fluorophore 2 with a large stokes shift of 117 nm and a 148-fold signal enhancement; and showed its applicability in detecting intracellular thiols in HeLa cells. This is a very interesting study that adds to our knowledge of designing fluorescent probes as sensors for intracellular biomolecules and metabolites. This will benefit the readership of Molecules and I recommend that the manuscript be published if the following points can be addressed by the authors:
1. The manuscript needs extensive editing of the English language.
2. The significance of this study relies on ultimately developing a clinical assay for quantifying changes in levels of specific thiols in the body. The authors should include a statement in the introduction about what is currently being used in the clinic if there is any, or a lack thereof.
3. In figure 4, how many measurements were done? Error bars should be included.
4. In figure 4A, the labels on the X axis can be replaced with the one-letter or three-letter abbreviations of the amino acid/thiol for easier reading.
5. In figure 6, the cells do not exhibit the normal morphology of HeLa cells. Is this an effect of treatment? It appears that the cells are adopting a circular shape that may indicate the loss of capacity to adhere to the glass/plastic surface. Did the authors measure the toxicity of probes 1 and 2 in HeLa cells? If not, it would be important to determine whether the probe or its cleavage products cause toxicity in cells.
6. Also in figure 6, the fluorescent signal is not uniformly distributed across the cytosol, which is not consistent with the distribution of thiols inside cells. Is there a specific compartment within the cell where the dye molecules preferentially accumulate? If yes, the authors should use a localization marker and show co-localization.
7. How did the authors determine the time point at which they imaged the cells after treatment with probe 1? Considering that the most abundant thiol in the cell is glutathione, how does the in vitro kinetics translate to the cellular system?
8. In page 2, line 71 “Figure S12” should be “Figure S11”
9. In page 2, lines 69-71 and figure S11 in the supporting information, are the other species in the reaction mixture (excess cysteine and other cleavage products) also present in the HRMS spectra, if yes, these should be labeled in the figure. Alternatively, the reaction mixture can be monitored by NMR to show the formation of the cleavage products.
10. In page 7, lines 190-201 should be removed from the manuscript.
Author Response
Dear Editor and Reviewers:
Thank you very much for your consideration on our manuscript! We really appreciate the time and effort that the reviewers have expended on our behalf. Their constructive suggestions and comments have undoubtedly resulted in an improved manuscript. We have revised the entire manuscript based on the Editor and Reviewers’ comments. Below are our point-by-point responses to the questions/suggestions raised by the reviewers.
Reviewers’ comments
Reviewer 1
Ma et al. describes a turn-on fluorescent probe for detecting thiols inside cells based on a thiol-induced cleavage of 2,4-dinitrobenzenesulfonate from a green-emitting phenothiazine benzothiazole dye. The authors showed that probe 1 can react with biothiols (cysteine, homocysteine, and glutathione) generating a green fluorophore 2 with a large stokes shift of 117 nm and a 148-fold signal enhancement; and showed its applicability in detecting intracellular thiols in HeLa cells. This is a very interesting study that adds to our knowledge of designing fluorescent probes as sensors for intracellular biomolecules and metabolites. This will benefit the readership of Molecules and I recommend that the manuscript be published if the following points can be addressed by the authors:
Comment 1: The manuscript needs extensive editing of the English language.
Response: According to the reviewer’s suggestion, we carefully checked the whole manuscript and corrected the typos and grammar mistakes.
Comment 2: The significance of this study relies on ultimately developing a clinical assay for quantifying changes in levels of specific thiols in the body. The authors should include a statement in the introduction about what is currently being used in the clinic if there is any, or a lack thereof.
Response: According to the reviewer’s suggestion, we added a statement in the introduction about what is currently being used in the clinic. ( In page 1, lines 35-37).
Comment 3: In figure 4, how many measurements were done? Error bars should be included.
Response: According to the review’s suggestion, we have added error bar in Figure 4. The error bar represents the standard deviation of three independent experiments.
Figure 4. (A) The fluorescence intensity at 530 nm of probe 1 (5.0 μM) upon the addition of the various common amino acid (from left to right: 100.0 μM for Ala, Asn, Arg, Asp, Gly, Gln, Glu, Lys, Leu, Met, Phe, Pro, Trp, Tyr, Thr, Ser, His, Ile, Val, GSH, Hcy, Cys). (B) The fluorescence intensity at 530 nm of probe 1 (5.0 μM) to Cys (100.0 μM) with the competition amino acid in PBS buffer. The error bar represents the standard deviation of three independent experiments (λexc = 413 nm, and t = 15 min).
Comment 4: In figure 4A, the labels on the X axis can be replaced with the one-letter or three-letter abbreviations of the amino acid/thiol for easier reading.
Response: According to the reviewer’s suggestion, we replaced with the three-letter abbreviations of the amino acid/thiol in figure 4A.
Comment 5: In figure 6, the cells do not exhibit the normal morphology of HeLa cells. Is this an effect of treatment? It appears that the cells are adopting a circular shape that may indicate the loss of capacity to adhere to the glass/plastic surface. Did the authors measure the toxicity of probes 1 and 2 in HeLa cells? If not, it would be important to determine whether the probe or its cleavage products cause toxicity in cells.
Response: The referee’s suggestion is really good. The result of the fluorescence imaging in HeLa cells is not very perfect, a better result should be given to verify probe 1 has potential applications for monitoring the thiols in vivo. According to the review’s suggestion, the fluorescence microscopy experiments were carried out in A549 cells. As is depicted in Figure 6, the A549 cells were stained with probe 1 solely (5.0 μM), and green fluorescence signal was appeared inside the cells (d, e and f). In sharp contrast, when the A549 cells were pretreated with NEM (1.0 mM) for 30 min to diminish the interference from intracellular biothiols, and then subsequently stained with probe 1. There was almost no remarkable fluorescence after the above treatment (a, b and c). All these results clearly revealed that probe 1 was cell membrance permeable and could be used to visualize thiols in living cells.
Figure 6. Confocal fluorescence contrast images: bright image (a, d) fluorescence image (b, e) and overlay image (c, f). Top row: living A549 cells pretreated with 1.0 mM NEM and then incubated with probe 1; bottom row: living A549 cells only incubated with probe 1 for 30 min at 37 oC.
According to the reviewer’s suggestion, we performed MTT assays on probe 1 and dye 2 to evaluate its cytotoxicity to A549 cells. As shown in Fig. S14 and S15, probe 1 and dye 2 had a low cytotoxicity to living cells.
Fig. S12 Percentage of viable A549 cells after treatment with different concentrations of probe 1 for 24 hours.
Fig. S13 Percentage of viable A549 cells after treatment with different concentrations of dye 2 for 24 hours.
Comment 6: Also in figure 6, the fluorescent signal is not uniformly distributed across the cytosol, which is not consistent with the distribution of thiols inside cells. Is there a specific compartment within the cell where the dye molecules preferentially accumulate? If yes, the authors should use a localization marker and show co-localization.
Response: The referee’s suggestion is really good. The result of the fluorescence imaging in HeLa cells is not very perfect, a better result should be given to verify probe 1 has potential applications monitor the thiols in vivo. According to the review’s suggestion, the fluorescence microscopy experiments were carried out in A549 cells (Fig.6). All these results clearly revealed that probe 1 was cell membrance permeable and could be used to visualize thiols in living cells.
Comment 7: How did the authors determine the time point at which they imaged the cells after treatment with probe 1? Considering that the most abundant thiol in the cell is glutathione, how does the in vitro kinetics translate to the cellular system?
Response: Time-dependent fluorescence response of probe 1 with Cys, Hcy and GSH were investigated respectively. As depicted in Fig. 5A, the reaction of probe 1 with thiols is almost finished within 15 min. The reaction rates of probe 1 with GSH (100.0 µM) and Hcy (100.0 µM) are much slower than that for Cys (100.0 µM), which may be attributed to the steric hindrance of Hcy and GSH. The physiological concentrations of thiols were Cys (30.0−200.0 μM), Hcy (5.0-13.9 μM) and GSH (1 mM), respectively (Sensors and Actuators B: Chemical, 2017, 253, 400-406). It is well known that the higher concentration of GSH resulted in a faster reaction. On the basis of the above principle, 30 min was selected as the reaction time for the whole cell experiments.
Comment 8: In page 2, line 71 “Figure S12” should be “Figure S11”
Response: According to the reviewer’s suggestion, we rectified the error.
Comment 9: In page 2, lines 69-71 and figure S11 in the supporting information, are the other species in the reaction mixture (excess cysteine and other cleavage products) also present in the HRMS spectra, if yes, these should be labeled in the figure. Alternatively, the reaction mixture can be monitored by NMR to show the formation of the cleavage products.
Response: According to the reviewer’s suggestion, we have not examined the cleavage products in the HRMS spectra. However, thiol-induced cleavage of intramolecular fluorescence quenching agent, such as 2,4-dinitrobenzenesulfonate or 2, 4-dinitrobenzenesulfonamide moieties, has been proven to be especially effective; This reaction mechanism has been proven by Maeda and co-workers in 2005 (Angew. Chem. Int. Ed. 2005,44,2922-2925). Moreover, this strategy has been demonstrated with great success using a number of prototypical fluorescent chromophores such as fluorescein, rhodamine, BODIPY, benzothiazole, resorufin and other species (Bioorg. Med. Chem. Lett., 2008, 18, 2246–2249; Biosens. Bioelectron., 2011, 26, 3012–3017;Org.Biomol. Chem., 2011, 9, 3844–3853; Org. Biomol. Chem., 2010, 8, 3627–3630;Org. Biomol.Chem., 2012, 10, 1966–1968; Tetrahedron Lett., 2012, 53, 2332–2335; Analyst, 2011,136, 191–195; Org. Lett., 2008, 10,37–40;Org. Biomol. Chem., 2009, 7, 4017–4020). Based on the above references, we used HRMS analysis to confirm the reaction mechanism.
Comment 10: In page 7, lines 190-201 should be removed from the manuscript.
Response: The referee’s suggestion is really good. We deleted the sentences (In page 7, lines 190-201).
Your help will be really appreciated.
Thank you very much in advance for your careful consideration!
Best wishes,
Yuanqiang Hao
Henan Key Laboratory of Biomolecular Recognition and Sensing, College of Chemistry and Chemical Engineering, Shangqiu Normal University, Shangqiu 476000, China
E-mail: haoyuanqiang@aliyun.com
Lin Liu
Key Laboratory of New Optoelectronic Functional Materials (Henan Province), College of Chemistry and Chemical Engineering, Anyang Normal University, Anyang 455000, China
E-mail: liulin@aynu.edu.cn

Reviewer 2 Report
Recommendation: The manuscript may be publishable after a major revision.
The manuscript by Hao Y. and Diu L. deals with a green-emitting fluorescent probe for imaging biothiols in living cells. The development of fluorescent probes for biothiols is widely explored because these compounds have numerous biological functions in living matter. The authors proposed a membrane-permeable chemodosimeter for biothiols. The work is carefully performed but some information is still missing.
Comments:
1.The authors should add the table with comparison of this dosimeter (detection limits, stock shifts, incubation time) to previously reported compounds which operate by using the same or similar chemical reactions.
2. The authors should explain whether pH = 7.4 was observed for PBS buffer before or after addition of DMSO.
3. Please, add the references proving that the non-emissive character of compound 1 in aqueous media results from PET process (Page 2, line 64).
4. Page 2, lines 67-71. Please, reformulate this discussion. In fact, detection of compound 2 by HRMS analysis proves the mechanism of the sensing process. "Cys-sensing behavior of compound 1" is not a proof of the mechanism.
5. Please, explain the accumulation of green color on the surface of HeLa cells (figure 6, f). It seems that the membrane permeability of chemosensor 1 is quite low.
6. The structure of compounds 1 and 2 does not correspond to their 13C NMR spectra. Please, add the intensity of the signals.
7. In 1H NMR spectra of compounds 1 and 2, NCH2CH2CH2 protons should have a more complicated pattern than singlets and doublets. Please, check these data or recorded the spectra at different temperature.
8. The purity of compound 1 is doubtful due to a low intensity of the signals in 13C NMR spectrum and broad signals in 1H NMR spectrum. Please, prove the purity of this compound improving NMR spectra.
Minor remarks:
1. Abstract - Please, reformulate sentence 2 (lines 15-17).
2. Page 2, line 55 – Please, delete "2. Results"
3. Scheme 1. Please, change "buffer" by "PBS buffer".
4. Page 2, line 60- Please, change "were" by "are".
5. Page 2, line 66. – Please, change "under this condition" by "under these conditions".
6. Page 2, line 67 – Please, change "maxem" by "max"
7. Figure1.Title – Please, add lexc = , and t = 15 min
8. Figure 2. Title - Please, add the incubation time and add lexc =…
9. Figures1-4 – Please, add units (a.u.) for the Y axe.
10. Page 7, lines190-201 –Please, delete these author's rules.
11. Page 7, line 204 – Please, change "displayed" by "displays".
12. Page 7, line 208. Supplementary materials – Please, delete the line 208 containing the rules for authors.
13. References. Please, check all abbreviations for journals.
14. Suplementary materials. Please, complete the titles of all figures by full experimental details (lexc = , and t =….). Do the experiments were carried out in PBS buffer or in a mixture of PBS buffer and 20% DMSO ?
Author Response
Dear Editor and Reviewers:
Thank you very much for your consideration on our manuscript! We really appreciate the time and effort that the reviewers have expended on our behalf. Their constructive suggestions and comments have undoubtedly resulted in an improved manuscript. We have revised the entire manuscript based on the Editor and Reviewers’ comments. Below are our point-by-point responses to the questions/suggestions raised by the reviewers.
Reviewers’ comments
Reviewer 2
The manuscript by Hao Y. and Diu L. deals with a green-emitting fluorescent probe for imaging biothiols in living cells. The development of fluorescent probes for biothiols is widely explored because these compounds have numerous biological functions in living matter. The authors proposed a membrane-permeable chemodosimeter for biothiols. The work is carefully performed but some information is still missing.
Comment 1: The authors should add the table with comparison of this dosimeter (detection limits, stock shifts, incubation time) to previously reported compounds which operate by using the same or similar chemical reactions.
Response: According to the review’s suggestion, we have presented a comparative account of thiols sensing in living cells from the large number of published papers (Table S1). As shown in Table S1, probe 1 showed larger stokes shift, relatively lower detection limit and shorter response time compared to other thiol sensors.
Table S1. Comparison of fluorescent probes for biothiols.
Probes | λex/λem (nm) | Stokes shift (nm) | Limit of detection | Response time | Reference |
450/540 | 90 | 1.5×10-8M | 10 min | Sensors actuat B-Chem, 2016, 223, 274-279.
| |
454/521 | 67 | 0.16 μM | 10 min |
Tetrahedron Letters, 2016, 57, 2478-2483 | |
280/482 | 202 | 2.0×10-8M | 20 min |
Tetrahedron, 2017, 73, 589-593 | |
309/510 | 201 | 0.17 μM | 10 min | Tetrahedron Letters, 2017, 58, 2654-2657.
| |
370/464 | 94 | 4.11×10-7M | 12 h | Analyst, 2013, 138, 7169-7174. | |
353/450 | 97 | 30 nM | 2 h | Chem Commun, 2013, 49,4640-4642. | |
413/530 | 117 | 0.12 μM | 15 min | This work |
Comment 2: The authors should explain whether pH = 7.4 was observed for PBS buffer before or after addition of DMSO.
Response: In the manuscript, place 5.0 μM of probe diluted solution mixed with appropriate analytes stock in 3.0 mL test cell (7.4 PBS buffer containing 20% DMSO), shake well and incubate 15 min each time at room temperature to make spectra measurements. According the references (Tetrahedron 73 (2017) 589-593; Journal of Photochemistry & Photobiology A: Chemistry 363 (2018) 7–12), the small amount of DMSO played the role of assisted dissolution. The pH of PBS did not be changed.
Comment 3: Please, add the references proving that the non-emissive character of compound 1 in aqueous media results from PET process (Page 2, line 64).
Response: The referee’s suggestion is really good. According to the review’s suggestion, we cited the literature as reference 44 in the revised manuscript.
Comment 4: Page 2, lines 67-71. Please, reformulate this discussion. In fact, detection of compound 2 by HRMS analysis proves the mechanism of the sensing process. "Cys-sensing behavior of compound 1" is not a proof of the mechanism.
Response: According to the review’s suggestion, we had reformulated this discussion.
Comment 5: Please, explain the accumulation of green color on the surface of HeLa cells (figure 6, f). It seems that the membrane permeability of chemosensor 1 is quite low.
Response: The referee’s suggestion is really good. The result of the fluorescence imaging in HeLa cells is not very perfect, a better result should be given to verify probe 1 has potential applications monitor the thiols in vivo. According to the review’s suggestion, the fluorescence microscopy experiments were carried out in A549 cells (Fig.6). All these results clearly revealed that probe 1 was cell membrance permeable and could be used to visualize thiols in living cells.
Comment 6: The structure of compounds 1 and 2 does not correspond to their 13C NMR spectra. Please, add the intensity of the signals.
Response: According to the review’s suggestion, we have confirmed the structures of compound 1 and 2 with 13C MR and added the intensity of the signals in Figure S9 and S13.
Comment 7: In 1H NMR spectra of compounds 1 and 2, NCH2CH2CH2 protons should have a more complicated pattern than singlets and doublets. Please, check these data or recorded the spectra at different temperature.
Response: According to the review’s suggestion, we performed NMR spectral analysis again. Unfortunately, similar results were obtained. Maybe reason was the poor solubility of compounds in DMSO. Furthermore, the HRMS analysis also confirmed the structures of compounds 1 and 2.
Comment 8: The purity of compound 1 is doubtful due to a low intensity of the signals in 13C NMR spectrum and broad signals in 1H NMR spectrum. Please, prove the purity of this compound improving NMR spectra.
Response: To confirm the purity of probe 1, high performance liquid chromatography (HPLC) analysis was carried out. As illustrated in Fig. R1, probe 1 has good purity.
Figure R1. Conditions: eluent, CH3CN/H2O (v/v, 4/6); flow rate, 1.0 mL min-1; temperature, 25 ℃; detection wavelength, 400 nm; injection volume, 15.0 μL.
Minor remarks:
1. Abstract - Please, reformulate sentence 2 (lines 15-17).
Response: According to the review’s suggestion, we had reformulate sentence 2
2. Page 2, line 55 – Please, delete "2. Results"
Response: According to the review’s suggestion, we deleted "2. Results"
3. Scheme 1. Please, change "buffer" by "PBS buffer".
Response: According to the review’s suggestion, we changed "buffer" by "PBS buffer".
4. Page 2, line 60- Please, change "were" by "are".
Response: According to the review’s suggestion, we changed "were" by "are".
5. Page 2, line 66. – Please, change "under this condition" by "under these conditions".
Response: According to the review’s suggestion, we changed " under this condition" by "under these conditions".
6. Page 2, line 67 – Please, change "maxem" by "max"
Response: According to the review’s suggestion, we changed "maxem" by "max".
7. Figure1.Title – Please, add lexc = , and t = 15 min
Response: According to the review’s suggestion, we added them.
8. Figure 2. Title - Please, add the incubation time and add lexc =…
Response: According to the review’s suggestion, we added them.
9. Figures1-4 – Please, add units (a.u.) for the Y axe.
Response: The data of emission spectra were carried out using RF5301PC instrument from Shimadzu. The units are not (a.u.).
10. Page 7, lines190-201 –Please, delete these author's rules.
Response: According to the review’s suggestion, we deleted the author's rules.
11. Page 7, line 204 – Please, change "displayed" by "displays".
Response: According to the review’s suggestion, we changed "displayed" by "displays".
12. Page 7, line 208. Supplementary materials – Please, delete the line 208 containing the rules for authors.
Response: According to the review’s suggestion, we deleted the line 208
13. References. Please, check all abbreviations for journals.
Response: According to the review’s suggestion, we checked all abbreviations for journals.
14. Suplementary materials. Please, complete the titles of all figures by full experimental details (lexc = , and t =….). Do the experiments were carried out in PBS buffer or in a mixture of PBS buffer and 20% DMSO ?
Response: According to the review’s suggestion, we completed the titles of all figures by full experimental details (lexc = , and t =….). The experiments were carried out in a mixture of PBS buffer and 20% DMSO.
Your help will be really appreciated.
Thank you very much in advance for your careful consideration!
Best wishes,
Yuanqiang Hao
Henan Key Laboratory of Biomolecular Recognition and Sensing, College of Chemistry and Chemical Engineering, Shangqiu Normal University, Shangqiu 476000, China
E-mail: haoyuanqiang@aliyun.com
Lin Liu
Key Laboratory of New Optoelectronic Functional Materials (Henan Province), College of Chemistry and Chemical Engineering, Anyang Normal University, Anyang 455000, China
E-mail: liulin@aynu.edu.cn

Reviewer 3 Report
Ma et al. reports a new benzothiazole derivative which shows fluorescence turn-on effect upon reacting with biothiols. They studied the selectivity and sensitivity of the probe with Cys, Hcy and GSH. They also demonstrated the use of the probe for cell imaging. To be accepted for publication, the authours should address the following issues:
1. Owing to the important role of biothiols, new probes with better performance are always desirable. Despite of presenting all the experimental results, the authors should comment whether their probe has any advantages over the previous ones or not.
2. What is the concentration range of intracellular biothiols in physiological condition? The detection range should be relevant to that.
3. The cell morphology in Figure 6 indicates the HeLa cells are not very healthy. The cytotoxicity of the probe should be provided. Meanwhile, most of the fluorescence signals came from the intracellular region, which might indicate the cell permeability of the dye is poor. Prolonging the incubation time might help the internalisation.
4. More references highly relevant to the topic should be included. For example, Sens Actuat B-Chem, 2017, 252: 712; Science China Chemistry 2018, 882; Principles and Applications of Aggregation-Induced Emission, Springer, 391-407, et al.
5. Some of the paragraphs are from the template. Please delete them. E.g. page 7.
Author Response
Dear Editor and Reviewers:
Thank you very much for your consideration on our manuscript! We really appreciate the time and effort that the reviewers have expended on our behalf. Their constructive suggestions and comments have undoubtedly resulted in an improved manuscript. We have revised the entire manuscript based on the Editor and Reviewers’ comments. Below are our point-by-point responses to the questions/suggestions raised by the reviewers.
Reviewers’ comments
Reviewer 3
Ma et al. reports a new benzothiazole derivative which shows fluorescence turn-on effect upon reacting with biothiols. They studied the selectivity and sensitivity of the probe with Cys, Hcy and GSH. They also demonstrated the use of the probe for cell imaging. To be accepted for publication, the authours should address the following issues:
Comment 1: Owing to the important role of biothiols, new probes with better performance are always desirable. Despite of presenting all the experimental results, the authors should comment whether their probe has any advantages over the previous ones or not.
Response: According to the review’s suggestion, we have presented a comparative account of thiols sensing in living cells from the large number of published papers (Table S1). As shown in Table S1, probe 1 showed larger stokes shift, relatively lower detection limit and shorter response time compared to other thiol sensors.
Comment 2: What is the concentration range of intracellular biothiols in physiological condition? The detection range should be relevant to that.
Response: The physiological concentrations of thiols were Cys (30.0−200.0 μM), Hcy (5.0-13.9 μM) and GSH (1 mM), respectively (Sensors and Actuators B: Chemical, 2017, 253, 400-406). As seen in Figure 3, a good linear relationship (y = 12.5969 + 20.2985x, R2 = 0.9972) between fluorescence intensity at 530 nm and the Cys concentrations (0.0 ~ 10 µM) was obtained. The detection limit of probe 1 for Cys was calculated to be 0.12 μM on the basis of S/N = 3. Additionally, similar results were obtained affording 0.14 μM and 0.21 μM detection limits for Hcy and GSH, respectively (Figure S1-S4). The results demonstrated that probe 1 could be used as a turn-on fluorescent indicator for biothiols in aqueous media.
Comment 3: The cell morphology in Figure 6 indicates the HeLa cells are not very healthy. The cytotoxicity of the probe should be provided. Meanwhile, most of the fluorescence signals came from the intracellular region, which might indicate the cell permeability of the dye is poor. Prolonging the incubation time might help the internalisation.
Response: The referee’s suggestion is really good. The result of the fluorescence imaging in HeLa cells is not very perfect, a better result should be given to verify probe 1 has potential applications monitor the thiols in vivo. According to the review’s suggestion, the fluorescence microscopy experiments were carried out in A549 cells (Fig.6). All these results clearly revealed that probe 1 was cell membrance permeable and could be used to visualize thiols in living cells.
Comment 4: More references highly relevant to the topic should be included. For example, Sens Actuat B-Chem, 2017, 252: 712; Science China Chemistry 2018, 882; Principles and Applications of Aggregation-Induced Emission, Springer, 391-407, et al.
Response: The references suggested by the referee are very important to our work. Therefore, we have added them in the revised manuscript (References:36-38).
Comment 5: Some of the paragraphs are from the template. Please delete them. E.g. page 7.
Response: The referee’s suggestion is really good. We deleted the sentences (In page 7, lines 190-201).
Your help will be really appreciated.
Thank you very much in advance for your careful consideration!
Best wishes,
Yuanqiang Hao
Henan Key Laboratory of Biomolecular Recognition and Sensing, College of Chemistry and Chemical Engineering, Shangqiu Normal University, Shangqiu 476000, China
E-mail: haoyuanqiang@aliyun.com
Lin Liu
Key Laboratory of New Optoelectronic Functional Materials (Henan Province), College of Chemistry and Chemical Engineering, Anyang Normal University, Anyang 455000, China
E-mail: liulin@aynu.edu.cn
Round 2
Reviewer 1 Report
The authors have sufficiently addressed my concerns and I recommend publication in Molecules after correcting these minor points:
Page 1, line 27: correct spelling of ‘cysteine’
page 1, line 27: change ‘low molecule weight’ to ‘low molecular weight’
page 3, line 82: change ‘the flourescent intensity’ to ‘the fluorescence intensity’
page 5, lines 134-135: change ‘in vivo’ to ‘in live cells’.
page 5, line 141: change ‘rapid visualize’ to ‘rapid visualization of’
Author Response
Dear Editor and Reviewers:
Thank you very much for your consideration on our manuscript! We really appreciate the time and effort that the reviewers have expended on our behalf. Their constructive suggestions and comments have undoubtedly resulted in an improved manuscript. We have revised the entire manuscript based on the Editor and Reviewers’ comments. Below are our point-by-point responses to the questions/suggestions raised by the reviewers.
Reviewers’ comments
Reviewer 1
The authors have sufficiently addressed my concerns and I recommend publication in Molecules after correcting these minor points:
Page 1, line 27: correct spelling of ‘cysteine’
page 1, line 27: change ‘low molecule weight’ to ‘low molecular weight’
page 3, line 82: change ‘the flourescent intensity’ to ‘the fluorescence intensity’
page 5, lines 134-135: change ‘in vivo’ to ‘in live cells’.
page 5, line 141: change ‘rapid visualize’ to ‘rapid visualization of’
Response: Thanks very much for the reviewer’s careful checking. And we have revised corrected all these typos according to the reviewers’ suggestions.
Your help will be really appreciated.
Thank you very much in advance for your careful consideration!
Best wishes,
Yuanqiang Hao
Henan Key Laboratory of Biomolecular Recognition and Sensing, College of Chemistry and Chemical Engineering, Shangqiu Normal University, Shangqiu 476000, China
E-mail: haoyuanqiang@aliyun.com
Lin Liu
Key Laboratory of New Optoelectronic Functional Materials (Henan Province), College of Chemistry and Chemical Engineering, Anyang Normal University, Anyang 455000, China
E-mail: liulin@aynu.edu.cn
Reviewer 2 Report
Recommendation: The manuscript may be publishable after a minor revision.
The manuscript by Yuanqiang Hao and co-workers was carefully revisited and completed. The reported chemodosimeter exibit a time response and detection limits which are similar to those of the early reported sensors based on the same chemical reaction. Regarding Stokes shifts of all known chemodosimeters, I think that there are no serious problems to solve. However, the studies reported in this manuscript are useful for further optimization of fluorescent probes for biothiols and are of interest for the researchers working in this field. The article can be published after minor corrections:
1. Comment 2 (1st revision):
The presence of 20% of DMSO certainly changes the pH of the buffered solution. Thus, the authors should reformulate their conditions: the measurements were performed in a buffered solution prepared by addition of 20 % of DMSO to PBS buffer (pH 7.4). The captions of the figures should be also changed because the measurements were carried out in the solution containing PBS buffer (pH = 7.4) and 20% of DMSO.
2. The results obtained with HeLa cells should be also shortly discussed in the manuscript and the images should be added in SI.
3. Maybe compound 2 was isolated as a hydrochloride. This could be a reason of its low solubility and the low resolution observed in 1H NMR spectrum of this compound. Please, add elemental analysis for this compound.
Author Response
Dear Editor and Reviewers:
Thank you very much for your consideration on our manuscript! We really appreciate the time and effort that the reviewers have expended on our behalf. Their constructive suggestions and comments have undoubtedly resulted in an improved manuscript. We have revised the entire manuscript based on the Editor and Reviewers’ comments. Below are our point-by-point responses to the questions/suggestions raised by the reviewers.
Reviewers’ comments
Reviewer 2
Ma et al manuscript by Yuanqiang Hao and co-workers was carefully revisited and completed. The reported chemodosimeter exibit a time response and detection limits which are similar to those of the early reported sensors based on the same chemical reaction. Regarding Stokes shifts of all known chemodosimeters, I think that there are no serious problems to solve. However, the studies reported in this manuscript are useful for further optimization of fluorescent probes for biothiols and are of interest for the researchers working in this field. The article can be published after minor corrections:
Comment 1: The presence of 20% of DMSO certainly changes the pH of the buffered solution. Thus, the authors should reformulate their conditions: the measurements were performed in a buffered solution prepared by addition of 20 % of DMSO to PBS buffer (pH 7.4). The captions of the figures should be also changed because the measurements were carried out in the solution containing PBS buffer (pH = 7.4) and 20% of DMSO.
Response: Thanks very much for the reviewer’s careful checking. And we have revised all these figure captions according to the reviewers’ suggestions.
Comment 2: The results obtained with HeLa cells should be also shortly discussed in the manuscript and the images should be added in SI.
Response: We have supplemented the fluorescence imaging results for HeLa cells in SI now. On page 5, lines 143-145, “The capability of probe 1 for imaging biothiols in Hela cells was further evaluated, and the results displayed a similar pattern with that of A549 cells (Figure S14). ”
Figure S15 Confocal fluorescence contrast images: bright image (a, d) fluorescence image (b, e) and overlay image (c, f). Top row: living HeLa cells pretreated with 1.0 mM NEM and then incubated with probe 1; bottom row: living HeLa cells only incubated with probe 1 for 30 min at 37 oC.
Comment 3: Maybe compound 2 was isolated as a hydrochloride. This could be a reason of its low solubility and the low resolution observed in 1H NMR spectrum of this compound. Please, add elemental analysis for this compound.
Response: Thanks very much for the reviewer’s constructive comments. During the synthesis of compound 2, the precipitated crude product has been dissolved in dichloromethane and washed with sodium carbonate solution before the flash chromatographic purification to ensure the removal of hydrochloride. Due to the limited equipments in our laboratory, we conducted the HPLC-MS tests instead of elemental analysis. Compound 2 was dissolved in a mixture of acetonitrile and H2O (1/1, v/v) and then subjected to HPLC-MS analysis using a Waters Acquity UPLC H-Class and Waters Xevo G2-S QTof™ mass spectrometer system (Waters, Milford, MA, USA). The seperation colume is a Waters ACQUITY BEH 2.1 × 50 mm C18 1.7 μm column. Flow rate was 0.5 mL/min. Eluent components were water contained water (A) and acetonitrile (B). The mobile phase gradient was as follows: the proportion of phase B increased from 50 to 100% in 4.0 min, and then to 50% in 5.0 min. The HPLC profile only displayed a single peak (Rt = 4.68 min) corresponding to the highest mass-to-charge ratio value of 405.1092 which is consistent with molecular structure of compound 2 (m/z calcd for [C23H20N2OS2 + H+]+: 405.1095).
Figure S14 HPLC-MS result of compound 2.
Your help will be really appreciated.
Thank you very much in advance for your careful consideration!
Best wishes,
Yuanqiang Hao
Henan Key Laboratory of Biomolecular Recognition and Sensing, College of Chemistry and Chemical Engineering, Shangqiu Normal University, Shangqiu 476000, China
E-mail: haoyuanqiang@aliyun.com
Lin Liu
Key Laboratory of New Optoelectronic Functional Materials (Henan Province), College of Chemistry and Chemical Engineering, Anyang Normal University, Anyang 455000, China
E-mail: liulin@aynu.edu.cn

Reviewer 3 Report
The authors add new results to answer my questions. I am happy for the manuscript to be accepted by Molecules.
Author Response
Dear Editor and Reviewers:
Thank you very much for your consideration on our manuscript! We really appreciate the time and effort that the reviewers have expended on our behalf. Their constructive suggestions and comments have undoubtedly resulted in an improved manuscript. We have revised the entire manuscript based on the Editor and Reviewers’ comments. Below are our point-by-point responses to the questions/suggestions raised by the reviewers.
Reviewers’ comments
Reviewer 3
Comments and Suggestions for Authors
The authors add new results to answer my questions. I am happy for the manuscript to be accepted by Molecules.
Response: Thanks very much for the reviewer’s careful cheching.
Your help will be really appreciated.
Thank you very much in advance for your careful consideration!
Best wishes,
Yuanqiang Hao
Henan Key Laboratory of Biomolecular Recognition and Sensing, College of Chemistry and Chemical Engineering, Shangqiu Normal University, Shangqiu 476000, China
E-mail: haoyuanqiang@aliyun.com
Lin Liu
Key Laboratory of New Optoelectronic Functional Materials (Henan Province), College of Chemistry and Chemical Engineering, Anyang Normal University, Anyang 455000, China
E-mail: liulin@aynu.edu.cn